# Light Field Intensification in Optical Films Induced by Intercoupling of Defects and Organic Contamination

**DOI:** 10.3390/mi13030387

**Published:** 2022-02-28

**Authors:** Xin Chen, Xiu-Lan Ling, Ji Liu, Xiao-Feng Liu

**Affiliations:** 1School of Information and Communication Engineering, North University of China, Taiyuan 030051, China; 15635530642@163.com (X.C.); liuji@nuc.edu.cn (J.L.); 2Key Laboratory of Material Science and Technology for High Power Lasers, Shanghai Institute of Optics and Fine Mechanics, Shanghai 201800, China; liuxiaofeng@siom.ac.cn

**Keywords:** optical thin film, defect, organic contamination, laser-induced damage

## Abstract

Based on the finite-difference time-domain method, light field intensification in optical films due to the intercoupling of defects and organic contamination was analyzed. The results show that the intercoupling between the defect and the organic contamination droplet leads to an increase in the local electric field and the coupling effect enhances with the decreasing distance between the defect and the organic contamination droplet and the increasing diameter of the organic contamination droplet. The coupling effect of the defect and the organic contamination layer depends on not only the thickness of the organic contamination layer but also the refractive index of the organic contamination layer. With the thickness and the refractive index of the organic contamination layer increasing, the peak value of the electric field decreases. This work deepens the physical understanding of the degradation mechanism of laser-induced damage in optical thin films used in vacuum.

## 1. Introduction

Nowadays, optical films are being widely used in vacuum environments due to the development of laser technology used in vacuum and space [1,2,3]. It is well known that the laser damage resistance of optical films is seriously degraded in vacuum environments, which has limited the further development of laser technology in vacuum or space [4,5,6]. Once optical films are damaged, the optical performance of the whole laser system will be hindered, which can lead to a severe chain reaction and ultimately cause the entire laser system to fail.

Laser-induced damage degradation in closed vacuum environments is mainly attributed to organic outgassing contaminants depositing on the surface of optical thin films [7,8,9]. It is believed that nano-defects caused by the fabrication of optical thin films are a potential source and they can significantly affect the laser damage susceptibility. Studies have also shown that the coupling effects of defects and organic contamination induce severe laser-induced thermal damage [10,11,12,13]. The laser damage process is the result of the coupling of multi-physical fields, and local light field intensification is the primary and first process and the premise of subsequent damage. Therefore, analyzing the coupling optical field enhancement effects of defects and organic contamination is helpful in understanding the laser damage mechanism of optical films and develop the technology to improve the anti-laser damage ability of optical films used in vacuum.

During the last two decades, TiO_2_ thin films, owing to their relatively large band, have become the preferred high-refractive-index material used in optical thin-film coatings, such as in high-reflecting mirrors, beam splitters, polarizers for high power, and high-energy laser applications with stringent requirements regarding laser-induced damage thresholds. So, it is necessary to study the light field enhancement of TiO_2_ optical films.

In this paper, a light field model has been presented to investigate the laser-induced damage of TiO_2_ optical films on a k9 substrate due to the intercoupling of defects and organic contamination. Due to the high-power or high-peak-intensity laser systems, such as the NIF devices, LMJ instruments, Litton Laser Systems Division (LLSD), and ICF systems operating at a fundamental frequency (wavelength of 1064 or 1053 nm), and their third harmonic lasers with a nanosecond pulse duration, we mainly considered the incident laser beam with a wavelength of 1064 nm. The local light field enhancement effects (LFEs) can be simulated by the finite-difference time-domain (FDTD) software because outgassing organic contaminations have two different adsorbing modes on the surface of the optical films: the organic contamination layer and the organic contamination droplet. Toluene organic pollutants were chosen for simulation because they are common outgassing organic contamination molecules in a closed vacuum laser system. Moreover, we mainly paid attention to transparent or weak absorption organic contamination for an irradiated laser, which is in accordance with the practical situations in vacuum laser systems. Therefore, we first studied the coupling effect of defects and organic contamination droplet on the optical field distribution of the optical film. After that, in the presence of the organic contamination layer, the dependence of the coupling light field enhancement was also analyzed. Our investigations provide a quantitative analysis of experimental data reported in previous works and deepens the physical understanding of the laser-induced degradation mechanism of optical thin films used in vacuum.

## 2. Theoretical Model

The intercoupling model of defect and organic contamination in the single-layer TiO_2_ thin film is illustrated in Figure 1. A single-layer TiO_2_ film with a thickness of 300 nm containing a spherical defect is located on the k9 substrate. The incident Gaussian profile laser beam with a wavelength of 1064 nm enters vertically into the surface of film along the *y*-axis direction, and the peak electric field intensity is set as E_0_ = 1 V/m (corresponding to the peak energy density of 10 J/cm^2^). All simulations in this article were completed by the finite-difference time-domain (FDTD) software by solving Maxwell equations based on rigorous electromagnetic field theory [14,15,16,17,18], and the simulation area was two-dimensional. On the lower boundary, a perfectly matched layer (PML) was set as the boundary condition, and the periodic boundary conditions were applied on the left and right boundaries. The coordinate system in Figure 1 was employed in the whole simulation process and in the presentation of results. Using this model, we simulated the light field distribution of the non-absorbing mono-layer TiO_2_ film with a refractive index of 2.1 due to the intercoupling of a defect and organic contamination.

## 3. Results and Discussions

### 3.1. The Effect of the Defect on Light Field Intensification 

As has been mentioned before, in the nanosecond pulse width region, laser-induced damage of dielectric thin films was mainly initiated by the nanometric defect [17]. In the case of high-power pulsed-laser optics applications, nanometric structural defects with a convex shape, caused during the fabrication of optical thin films, can be a main source of enhanced light field, leading to laser damage, which we mainly focused on in our simulation accordingly. These defects were oxide defects in terms of chemical nature due to the splashing of film materials during processing. Therefore, we firstly analyzed the local light field enhancement of TiO_2_ optical films induced by a single non-absorbing spherical defect with a diameter of 160 nm and a refractive index of 1.5. Figure 2 shows that in the optical film, the maximum electric field was located on the surface of the ideal TiO_2_ film layer, whereas the peak intensity of the electric field lodged on the boundary of the defect and a light field intensification as large as 1.5× occurred in contrast to an ideal TiO_2_ thin film. The results reveal that the defect in the film is equivalent to a micro-lens [18], which can focus the incident light inducing the local light field intensification in the film. The micro-lens effects induced by the defect not only lead to a local increase in the light intensity in the defect and the surrounding film but also distort the light field distribution in the film.

### 3.2. The Coupling Effect of the Defect and the Organic Contamination Droplet 

Figure 3 denotes the coupling effect of a defect and an organic contamination droplet with a size of 200 nm and a refractive index of 1.46 on the local light field of a TiO_2_ optical film. We can see that the coupling of the defect and the organic contamination droplet induces a higher local light field enhancement in the TiO_2_ optical film and the peak intensity of the electric field lodged on the boundary of the defect and the organic contamination droplet, respectively. Moreover, it can be seen that for a defect of the same size and refractive index, the peak electric field is different when the distance between the defect and the organic contamination droplet is different; when the defect is in contact with the droplet, the peak electric field induced is the largest. The greater the distance, the lower the peak electric field induced. Therefore, the defects located near the surface of the film are more likely to intercouple with an organic contamination droplet to cause damage.

In addition, we investigated light field variation with organic contamination droplets of different sizes. In the simulation, three different refractive index values of the defect (1.5, 2.6, and 2.1) were selected, which are representative because the refractive index of the host material (TiO_2_) is 2.1. The results are presented in Figure 4. They reveal that the larger organic contamination droplets are more likely to couple with the defect and induce damage.

Therefore, organic contamination droplets adsorbed on the surface of a film will cause further local light field enhancement, in contrast to a clean film, which in turn intensifies the laser damage of the film. This result is also in agreement with the experimental data reported in the literature [10]. 

### 3.3. The Coupling Effect of the Defect and the Organic Contamination Layer 

#### 3.3.1. The Different Thicknesses of the Organic Contamination Layer 

Figure 5 shows the coupling effect of a defect and an organic contamination layer on the local light field of a TiO_2_ optical film when the thickness of the organic contamination layer is 20, 40, 60, 80, and 100 nm. In this simulation, the refractive index of the organic contamination layer is set to 1.46. It can be seen that the different thicknesses of the organic contamination layer induce different peak electric fields. With the thickness of the organic contamination layer increasing, the peak value of the electric field decreases and the more the defect is exposed on the surface, the more significant the focusing effect on the incident light is, so the local peak electric field induced is larger. So the organic contamination layer suppresses the focusing effect of the incident light caused by defects. Therefore, the decreasing laser-induced damage threshold of optical films used in vacuum is not attributed to the intercoupling effects between the defect and the organic contamination layer. This result is in accordance with our previous conclusions [10,12].

#### 3.3.2. Different Refractive Index Effects of the Organic Contamination Layer 

Our previous studies have shown that different organic contaminations cause different damage thresholds of an optical film [9]. For this reason, the coupling effect of a defect and an organic contamination layer with different refractive indexes on the optical field is simulated, as shown in Figure 6. It can be seen that for a defect lodged in the same position in an optical film, with the same refractive index, and of the same size, when the refractive index of the organic contamination layer is different, the peak electric field induced is different. An organic contamination layer with a high refractive index induces a lower peak value of the optical field. This result indicates that as far as possible, packaging organic materials with a high refractive index should be selected to use in a vacuum laser system.

## 4. Conclusions

Based on electric field modeling, the light field enhancement caused by the coupling of defects and organic contamination in a single-layer TiO_2_ film was calculated by the finite-difference time-domain method. The results show that the intercoupling between the defect and the organic contamination droplet leads to an increase in the local electric field and the coupling effect decreases with the increase of distance between the defect and the organic contamination droplet. At the same time, the larger organic contamination droplet is more likely to couple with the defect and damage the optical films. The coupling effect of the defect and the organic contamination layer depends on not only the thickness of the organic contamination layer but also the refractive index of the organic contamination layer. With the thickness and the refractive index of the organic contamination layer increasing, the peak value of the electric field decreases. The results are also in agreement with a bunch of experimental data reported in the literature. Our investigations further provide quantitative analysis of such data and deepens the physical understanding of the degradation mechanism of laser-induced damage in optical thin films used in vacuum.

## Figures and Tables

**Figure 1 micromachines-13-00387-f001:**
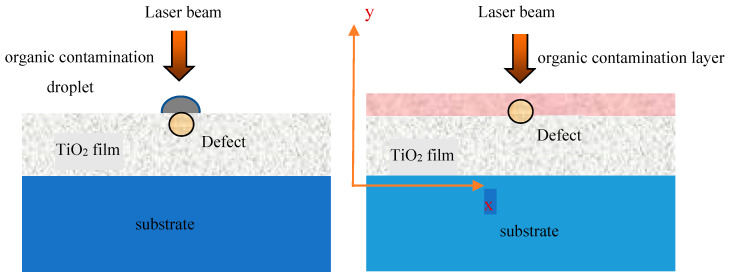
Intercoupling model of defect and organic contamination in the single-layer TiO_2_ thin film.

**Figure 2 micromachines-13-00387-f002:**
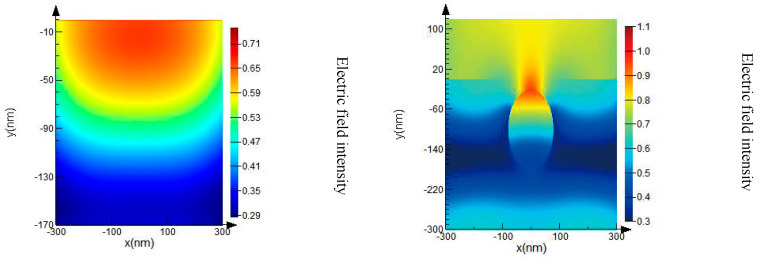
Light field distribution of films induced by the defect in the TiO_2_ optical film.

**Figure 3 micromachines-13-00387-f003:**
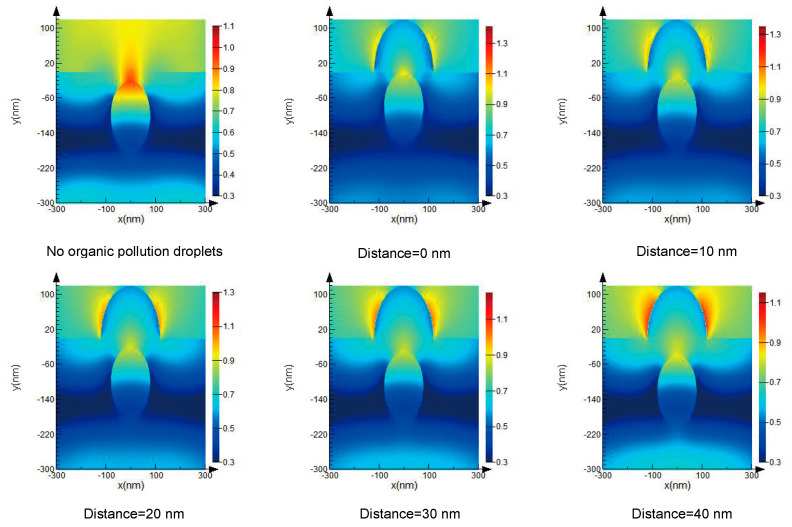
Light field enhancement with different coupling distances between a defect and an organic contamination droplet.

**Figure 4 micromachines-13-00387-f004:**
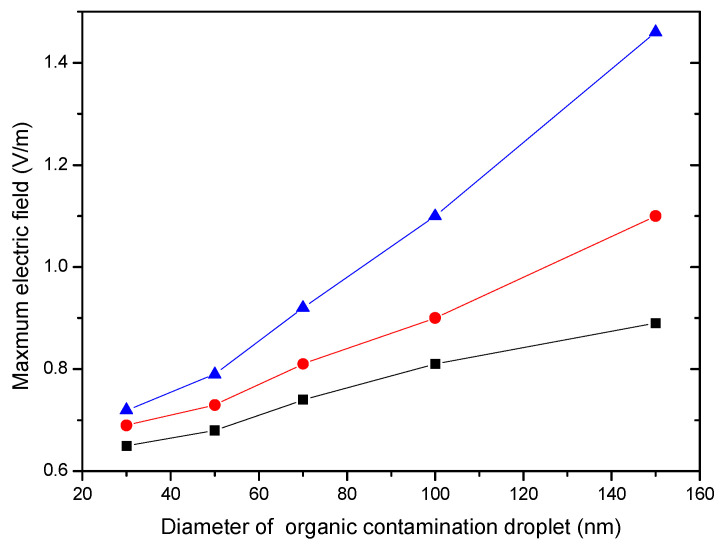
Light field variation with organic contamination droplets of different sizes. The blue, red, and black lines are the result of the different refractive indexes (1.5, 2.6, and 2.2, respectively) of the defect.

**Figure 5 micromachines-13-00387-f005:**
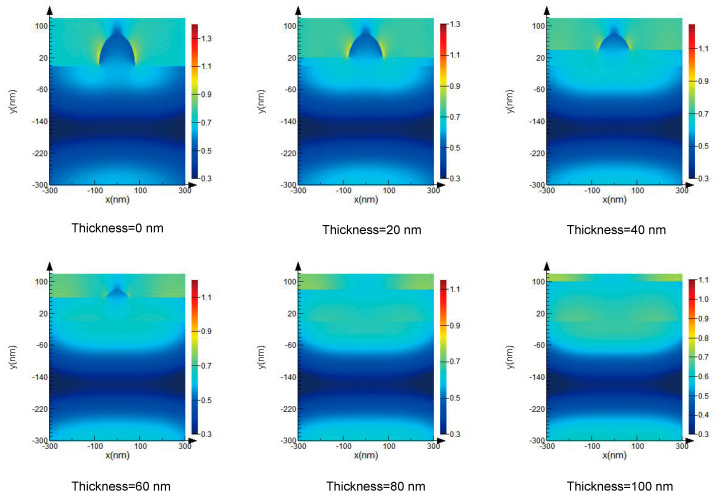
The coupling effect of defects and organic contamination layers of different thicknesses on the local light field of a TiO_2_ optical film.

**Figure 6 micromachines-13-00387-f006:**
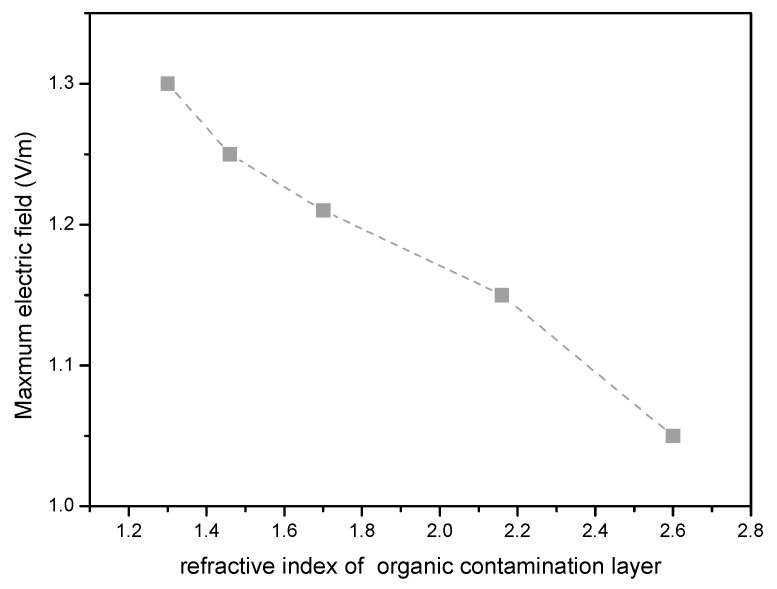
Light field variation with an organic contamination layer of different refractive indexes (the refractive index of the defect is 1.5).

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
