# Peer review of "Light Field Intensification in Optical Films Induced by Intercoupling of Defects and Organic Contamination"

_micromachines, 2022, doi:10.3390/mi13030387_

Round 1

Reviewer 1 Report

The paper treats a quite interesting subject which is of interest for designing laser systems (from laser beams sources to robots) used in a large number of technological applications.

As is proposed, the paper deals with the subject in an uncomplete way - to say the least-, and the authors look at the treated issues at a quite general level.  Directly  connected with this observations the question arises: why the article is considered by the authors suitable for publication in Micromachines journal.

This has to be mentioned not necessary in an explicit phrase, but outlining the applications in which the lasers and their optical components are used, both for developed of technologies on Earth or in space/microgravity.

As is the submitted paper may not be accepted for publication. It requires major improvements made to make its content more consistent.

Questions as mentioned below should receive some answers in the article that may be given if the authors decide to go ahead with the publication efforts.

  1. The authors speak about the contamination – either with thin films or with droplets- with organic compounds. They should mention first which are the support materials (which are coated with TiO2 thin layers) considered by them for the chosen optical components, and what components do they consider (totally reflecting mirrors, beam splitters, filters, polarizers, extraction windows, aso). They should also present and discuss about the particular organic compounds considered in the study. This particular information is presented at a too general level since, for instance, the absorption coefficients of the compounds and the light scattering (backward and/or forward) are important in the treated applications. Both categories of information are of importance in understanding the significance of the reported results.
  2. The authors should discuss about the particular defects, (dimensions micro- or nano-metric, chemical nature, aso) present in the optical materials used in the laser systems considered by them. The simple mention of refractive indices would not be enough.
  3. It is highly recommended that the authors consider and discuss the reason for which they have chosen 1064 nm near infrared beam, and the power levels of the laser beams considered in computations. A typical considered application is recommended to be described, even at a quite general level.
  4. As a concluding remark, it may be mentioned that the conclusions of the article are in agreement with a bunch of experimental data reported in the literature. The novelty of the article could consist in providing a quantitative analysis of such data. This should be mentioned somehow in the article’s body.
  5. The English grammar and typing has to be corrected

Reviewer 2 Report

In section 2: can you name the software program used in simulation?

In Figure 2: Please label the color scale (light intensity ...)

P3, line 98: Do you have any suggestion to overcome this problem of contamination near the surface ? (nanomaterial layer ?)

P5, line142: selected (with) high

In conclusion: the authors have not mentioned anything about experimental validation of the simulation? do they have any plan to do that in future?

Author Response

Dear reviewer2:

1)The software program used in simulation in section 2 is named FDTD which is mentioned in revised manuscript.

2)The color scale in Figure 2 have been labeled as “Electric field intensity” according to your suggestion in revised manuscript.

3)We think that the control of the adsorption of organic contamination molecules on the surface of optical thin films is of main practical and economical interest for future design. And adsorption effect is dependent on both organic contamination molecules and the identity of adsorbed optical surfaces.The optical thin films with surface electric field of zero have weak adsorption action for organic contamination molecules in contrast with that of nonzero surface electric field. So, it should be considered that optical film with the surface electric field of zero is designed and used in closed vacuum laser system as much as possible through the design of film system.

4)In P5, line142: we have corrected the sentence “selected (with) high” in revised manuscript according to your suggestion.

5)It is true that we have not mentioned anything about experimental validation of the simulation in conclusion. Part experimental validation can be seen in our previous work “Laser-induced thermal damage simulations of optical coatings due to inter-coupling of defect and organic contamination, IEEE Photonics Journal, 10(4):6100707(2018)”. And now we are making further progress in this part of the work.

Thanks very much for your positive comments on our manuscript.

Round 2

Reviewer 1 Report

The article may be accepted as is.